# Using an Unsupervised Clustering Model to Detect the Early Spread of SARS-CoV-2 Worldwide

**DOI:** 10.3390/genes13040648

**Published:** 2022-04-07

**Authors:** Yawei Li, Qingyun Liu, Zexian Zeng, Yuan Luo

**Affiliations:** 1Department of Preventive Medicine, Feinberg School of Medicine, Northwestern University, Chicago, IL 60611, USA; yawei.li@northwestern.edu; 2Department of Immunology and Infectious Diseases, Harvard T.H. Chan School of Public Health, Boston, MA 02115, USA; qingyunliu@hsph.harvard.edu; 3Department of Data Science, Dana Farber Cancer Institute, Harvard T.H. Chan School of Public Health, Boston, MA 02215, USA; zexian_zeng@dfci.harvard.edu

**Keywords:** deep learning clustering, population structure, evolution, nucleotide substitutions, SARS-CoV-2

## Abstract

Deciphering the population structure of SARS-CoV-2 is critical to inform public health management and reduce the risk of future dissemination. With the continuous accruing of SARS-CoV-2 genomes worldwide, discovering an effective way to group these genomes is critical for organizing the landscape of the population structure of the virus. Taking advantage of recently published state-of-the-art machine learning algorithms, we used an unsupervised deep learning clustering algorithm to group a total of 16,873 SARS-CoV-2 genomes. Using single nucleotide polymorphisms as input features, we identified six major subtypes of SARS-CoV-2. The proportions of the clusters across the continents revealed distinct geographical distributions. Comprehensive analysis indicated that both genetic factors and human migration factors shaped the specific geographical distribution of the population structure. This study provides a different approach using clustering methods to study the population structure of a never-seen-before and fast-growing species such as SARS-CoV-2. Moreover, clustering techniques can be used for further studies of local population structures of the proliferating virus.

## 1. Introduction

The COVID-19 pandemic, caused by severe acute respiratory syndrome coronavirus 2 (SARS-CoV-2) [1,2], has spread throughout the world since 2019. To understand the molecular characteristics of the virus, a large collaborative project known as the Global Initiative on Sharing All Influenza Data (GISAID) was developed to collect and publish the sequenced genomes of SARS-CoV-2 worldwide. The database has helped researchers in investigating comprehensive patterns of SARS-CoV-2 with high efficiency.

As a newly emerging pandemic virus, exploring the genetic diversity, evolutionary trajectory and possible routes of transmission of SARS-CoV-2 from its natural reservoir to humans is crucial for supporting clinical studies. For that reason, studies have implemented phylogenetic trees to investigate the evolutionary trajectory of SARS-CoV-2 and its strain diversification [3,4,5]. A phylogenetic tree is a graph showing the dissimilarities and distances between various biological entities based on their pairwise distance metrics [6,7]. Nevertheless, given the continuous accumulation of entities, the human mind is limited in effectively handling and fully utilizing the accumulation of such enormous amounts of entities through the phylogenetic tree. As a result, grouping similar entities into the same cluster and then extracting the major characteristics of each cluster greatly facilitated the efficiency of our analysis. Typically, entities can be handily grouped in terms of the distance matrix and the bifurcations between branches of leaves on the phylogenetic tree. However, when the entity size is enormous, manually partitioning the clades of the phylogenetic tree becomes much more challenging.

With the development of machine learning algorithms, machine learning-based approaches have played a critical role in phylogenetic analysis. Compared with other methods, machine learning-based methods can find a better balance between accuracy and running time [8]. To our knowledge, machine learning methods have been widely implemented in various aspects of phylogenetic analyses, including phylogenetic tree construction [8,9,10,11,12], ancestral relationship identification [13], evolutionary rate estimation [14,15], gene evolutionary mechanisms research [16], and population structure analysis [17]. The unsupervised clustering models, a subgroup of machine learning models, have acted as more productive and robust solutions to effectively and accurately cluster these entities. The objective of clustering is to simultaneously minimize intra-cluster distances and maximize inter-cluster distances [18]. Although finding the globally optimal solution is near-impossible, a valuable and effective clustering method allows us to easily extract the inner differences between clusters within the dataset and inform downstream analyses.

In this study, to elucidate the genetic structure and evolution of early SARS-CoV-2, we downloaded 16,873 genomes from the published database, and used a clustering model based on the recently published state-of-the-art deep embedding clustering (DEC) method [19] to cluster these genomes into six major clusters. Interestingly, the proportions of clusters across the six continents (Africa, Asia, Europe, North America, Oceania, and South America) displayed distinct geographical distributions. Our comprehensive analysis revealed that the unique geographical distributions across the clusters are influenced by both intrinsic genetic factors and the migration of humans. This study provides an example of using machine learning algorithms to assist phylogenetic analysis, supporting the exploration of the spread of SARS-CoV-2 across humans worldwide.

## 2. Materials and Methods

### 2.1. SARS-CoV-2 Sequencing Collection

The genomes of SARS-CoV-2 uploaded before 15 May 2020 were downloaded from GISAID (https://db.cngb.org/gisaid/, accessed on 15 May 2020). Only the strains marked as “high coverage” were maintained in our study. In terms of GISAID, the definition of “high coverage” was strains with <1% Ns and <0.05% unique amino acid mutations (not seen in other sequences in databases), and no insertion/deletion unless verified by the submitter. Moreover, non-human host strains and assemblies of total genome length less than 29,000 bps strains were removed. A total of 16,873 strains passed the filtration. The Accession IDs with demographic information for the strains are listed in Appendix A. The distribution of the number of collected strains per day is shown in Appendix A.

### 2.2. Mutation Calling and Phylogeny Reconstruction

We followed the Nextstrain pipeline [20] to map the downloaded SARS-CoV-2 strains to the reference genome (GenBank Accession Number: NC_045512.2). We used CLUSTALW 2.1 [21] for multiple and pairwise sequencing alignments. Considering putatively artefactual substitutions and gaps located at the beginning and end of the sequencing, we masked the first 130 bps and last 50 bps in the mutation calling step based on the Nextstrain pipeline [20].

To calculate the mutation rate of the strains, we defined the collection date of the reference genome as the reference date. The mutation rate (substitutions per year) of a given strain can be expressed as:$$\mathrm{mutation}\;\mathrm{rate}=\frac{\mathrm{mutation}\;\mathrm{counts}\;\left(\mathrm{nucleotide}\;\mathrm{substitations}\right)\times 365}{\mathrm{days}\;\mathrm{from}\;\mathrm{the}\;\mathrm{collection}\;\mathrm{date}\;\mathrm{of}\;\mathrm{the}\;\mathrm{given}\;\mathrm{strain}\;\mathrm{to}\;\mathrm{the}\;\mathrm{collection}\;\mathrm{date}\;\mathrm{of}\;\mathrm{the}\;\mathrm{reference}\;\mathrm{strain}}$$The average mutation rate was the mean mutation rate of all strains within a population.

A phylogenetic tree was generated using all 16,873 strains. We employed FastTree 2 [22] to reconstruct the phylogenetic tree of SARS-CoV-2 using the nucleotide substitutions of these strains. The phylogeny tree was rooted using FigTree v1.4.4 and visualized using the online tool Interactive Tree Of Life (iTOL v5) [23].

### 2.3. Feature Extraction and Data Clustering

We achieved a total of 7970 unique substitutions across 16,873 strains through mutation calling. Herein, we used the aggregated substitution profiles to train our unsupervised clustering model. The input of the clustering model was a matrix with dimensions 16,873 × 7970, where each row represented a strain, and each column represented a unique substitution. If a substitution “*s*” was detected in the strain “*i*”, the cell with the coordinate (*i*, *s*) was marked as “1”, whereas no mutation was marked as “0”.

We implemented a clustering model based on the recently published DEC model [19] to iteratively cluster the strains. We first used K-means clustering to initialize the centroids of clusters. To update the cluster assignments, we used Student’s t-distribution as a kernel to measure the distance ($q_{ij}$) of each strain ($h_{i}$) to each centroid ($u_{j}$):$$q_{ij}=\frac{{\left(1+||h_{i}-u_{j}{||}^{2}/\alpha \right)}^{-\frac{\alpha +1}{2}}}{\sum_{j^{\prime }=1}^{K}{\left(1+||h_{i}-u_{j^{\prime }}{||}^{2}/\alpha \right)}^{-\frac{\alpha +1}{2}}}$$

Based on the equation, the distance $q_{ij}$ can be interpreted as the probability of assigning strain *i* to cluster *j*. The α was the degree of freedom of the distribution. For convenience, we assumed that α=1 in this study. Next, we defined an auxiliary target distribution *P* by raising each $q_{ij}$ to the second power to enhance strains assigned with high confidence:$$p_{ij}=\frac{q_{ij}^{2}/\sum_{i=1}^{N}q_{ij}}{\sum_{j^{\prime }=1}^{K}\left(q_{ij^{\prime }}^{2}/\sum_{i=1}^{N}q_{ij^{\prime }}\right)}$$
where the denominator was used to normalize the loss contribution of each centroid to prevent large clusters from distorting the feature space. Finally, we defined the Kullback–Leibler (KL) divergence loss as the objective function:$$L=\mathrm{KL}(P||Q)=\sum_{i=1}^{N}\sum_{j=1}^{K}p_{ij}log\frac{p_{ij}}{q_{ij}}$$
where the parameters and cluster centroids were jointly optimized by minimizing L using the Stochastic Gradient Descent (SGD) with momentum. Notably, the DEC model needs to pre-specify the number of clusters. To this end, we tested different cluster numbers from 2 to 20 and plotted the sum of squared errors (SSE) and Bayesian information criterion (BIC) [24] curves. The number of clusters was determined using the elbow method [25]. Moreover, considering the possible overfitting, we tested the non-negative matrix factorization (NMF) method for dimensionality reduction [26,27,28]. However, the mean intra-cluster pairwise genetic distances indicated that directly using the entire substitution binary matrix as the input feature achieved a minimal mean pairwise genetic distance.

### 2.4. Pairwise Genetic Distance

Pairwise genetic distance is a measure of the genetic divergence between two entities. Herein, we used Hamming distance (*d*(*i*, *j*)) to measure the pairwise genetic distance between two strains (*i*, *j*) based on their nucleotide substitutions. Thus, the pairwise genetic distance is the number of nucleotide positions in which the two strains are different. For a cluster (*A*), *N(A)* denotes the number of strains of the cluster, and the mean intra-cluster pairwise genetic distance ($D{\left(A\right)}_{intra}$) can be computed as:$$D{\left(A\right)}_{intra}=\frac{2\times \sum_{i=1}^{N\left(A\right)-1}\sum_{j=i+1}^{N\left(A\right)}d\left(i,j\right)}{N\left(A\right)\times \left(N\left(A\right)-1\right)}$$Likewise, the mean inter-cluster pairwise genetic distance ($D{\left(A\right)}_{inter}$) is:$$D{\left(A\right)}_{inter}=\frac{\sum_{i=1}^{N\left(A\right)}\sum_{j=1}^{16,873-N\left(A\right)}d\left(i,j\right)}{N\left(A\right)\times \left(16,873-N\left(A\right)\right)}$$

In this study, we used the mean overall intra-cluster pairwise genetic distance ($D{\left(\mathrm{overall}\right)}_{intra}$) as the metric to evaluate the performance of a clustering model. For the six clusters, $D{\left(\mathrm{overall}\right)}_{intra}$ can be expressed as:$$D{\left(\mathrm{overall}\right)}_{intra}=\frac{2\times \sum_{k=1}^{6}\sum_{i=1}^{N\left(k\right)-1}\sum_{j=i+1}^{N\left(k\right)}d\left(i_{k},j_{k}\right)}{\sum_{k=1}^{6}N\left(k\right)\times \left(N\left(k\right)-1\right)}$$
where $i_{k}$ and $j_{k}$ denote the two strains in cluster *k*, respectively. In terms of the equation, a smaller $D{\left(\mathrm{overall}\right)}_{intra}$ represents better clustering performance.

### 2.5. Simpson’s Diversity Index

Simpson’s Diversity Index (D*_sim_*) is a measure of diversity that considers the number of entities as well as their abundance. The index measures the probability that two randomly selected individuals are the same. The formula to calculate the value of the index is:$$D_{sim}=1-\frac{\sum_{all\;traits}n\left(n-1\right)}{N\left(N-1\right)}$$
where 𝑛 is the number of individuals displaying one trait and 𝑁 is the total number of all individuals. The value of D*_sim_* ranges between 0 and 1. With this index, 1 represents infinite diversity and 0 denotes no diversity.

### 2.6. Inferring Positive/Purifying Selection of Individual Sites

To test which position was under selective pressure, we used a set of programs available in HyPhy to calculate nonsynonymous (dN) and synonymous (dS) substitution rates on a per-site basis to infer pervasive selection. Fast Unconstrained Bayesian AppRoximation (FUBAR) was applied to detect overall sites under positive selection. The positively selected sites were identified using a probability larger than 0.95 using the FUBAR method.

### 2.7. Pairwise Mutation Dependency Score

Pairwise mutation dependency scores can measure the order in which genetic mutations are acquired within a cluster. For two selected mutations *X* and *Y*, the score *S(X|Y)* represents the proportion of strains that accumulated *X* among the strains that accumulated mutation *Y*. *S(X|Y)* and *S(Y|X)* can be calculated using the following functions:$$S\left(X|Y\right)=\frac{\sum_{all\;samples}S_{X}=1\;\&\;S_{Y}=1}{\sum_{all\;samples}S_{Y}=1}$$
$$S\left(Y|X\right)=\frac{\sum_{all\;samples}S_{X}=1\;\&\;S_{Y}=1}{\sum_{all\;samples}S_{X}=1}$$
where $S_{X}=1$ denotes that the sequence has a mutation *X*. The pairwise mutation dependency score displays the correlation and the timescale relationship of the two mutations. The value of *S(X|Y)* and *S(Y|X)* ranges between 0 and 1. With this index, *S(X|Y)* = 1 with *S(Y|X)* < 1 represents that mutation *Y* occurs after mutation *X*. In contrast, *S(X|Y)* = 1 with *S(Y|X)* = 1 represents that the two mutations occur simultaneously and are genetically linked. Statistical analyses and data presentations were generated using Python 3.7.6.

### 2.8. Statistical Analysis and Data Visualization

For each country with SARS-CoV-2 data available, clustering proportions were calculated and plotted on the world map using the tool Tableau Desktop 2020.2. The T-distributed Stochastic Neighbor Embedding (t-SNE) plot was generated using the aggregated substitution profiles for the 16,873 strains. We used the sklearn.manifold.TSNE package in Python for t-SNE training and the seaborn package in Python for visualization. The other figures were plotted using the ggplot2 in R 4.0.1, the seaborn package in Python 3.7.6 and GraphPad Prism 8.0.2. Statistical analyses, and the clustering models were implemented in Python 3.7.6.

## 3. Results

### 3.1. Genetic Analysis Revealed High Genetic Diversity and Rapid Proliferation of SARS-CoV-2

We downloaded the whole genome sequencing of SARS-CoV-2 from GISAID, aligned the sequences, and called mutations. Across the 16,873 strains (Africa: 98, Asia: 1324, Europe: 9527, North America: 4765, Oceania: 1040 and South America: 119), we identified 7970 substitutions (4908 non-synonymous substitutions, 2748 synonymous substitutions, and 314 intronic substitutions), with 6.99 mutation counts per strain (Appendix A). In terms of the frequency spectrum of the mutations, we found that more than 99% of mutations had frequencies smaller than 1% (Appendix A). Of these mutations, 54.05% were singletons (can be observed in one strain) and 15.35% were doubletons (can be observed in two strains). The high proportions of these low-frequency mutations suggest a rapid proliferating pattern of SARS-CoV-2 [29]. Of the downloaded 16,873 strains, we identified a total of 8706 unique strains, 7078 of which were singletons (Appendix A). The Simpson’s diversity index was 0.8222, indicating that two random strains would have a high probability of being genetically different. In summary, the distributions of these mutations and strains demonstrated the high genetic diversity of SARS-CoV-2.

### 3.2. Clustering of SARS-CoV-2 Displayed Varied Proportions of the Clusters in Different Continents

There were 8706 unique strains within the entire SARS-CoV-2 population; thus, grouping these strains into multiple major subtypes was necessary to informatively formulate their population structure. Making use of the machine learning (ML) algorithms, we applied unsupervised ML models to cluster SARS-CoV-2 strains using their substitution profiles, as SARS-CoV-2 exhibited limited substitutions per strain as well as little ongoing horizontal gene exchange [4].

Our initial work [30] used an unsupervised clustering model based on DEC [19] (see Section 2), to group the strains into six major clusters. To test the accuracy and biological interpretation of the clustering results, we mapped the six clusters against the phylogenetic tree. Each cluster of the SARS-CoV-2 strains were compact in the phylogenetic tree (Figure 1A). In addition, the pairwise genetic distances between intra-clustering and inter-cluster and t-SNE plot demonstrated that these strains were adequately isolated between clusters (Figure 1B,C and Appendix A).

We investigated the distributions of clustering results across continents and found that the clusters diverged in their geographical distributions (Figure 2, Table 1). For example, European strains were the dominant strains in cluster A (81.92%), cluster C (71.97%), and cluster F (85.73%); North American strains were the major strains in cluster D (66.39%) and cluster E (61.88%); and Asian and European strains occupied 76.47% in cluster B.

It is noted that approximately 85% of the strains were collected from North America and Europe (Appendix A). Thus, directly comparing the proportion of continents in each cluster was not informative in terms of sampling bias. We next considered the distribution of clusters in each continent. Most continents were concentrated in only one or two clusters, including Africa (cluster C: 66%), Asia (cluster B: 49%), Europe (cluster C and cluster F: 64%), North America (cluster D plus cluster E: 74%), and South America (cluster C plus cluster F: 78%). In contrast, the distribution of the strains in Oceania was uniformly dispersed across the six clusters, suggesting that the diversity of SARS-CoV-2 was higher in Oceania than in the other continents. Note that, of all clusters, cluster C dispersed globally, and was the only cluster with frequencies exceeding 10% across all six continents, suggesting that cluster C might be more disseminated than the other clusters.

The different geographical distributions for the six clusters could be due to intrinsic genetic factors, extrinsic factors such as the migration of humans, or both. Hence, we next aimed to explore the genomic characteristics of these clusters, as well as the transmission and human migration of the virus across the globe.

### 3.3. The Genetic Variance Analyses Indicated High Diversity between Clusters

If the different geographical distributions for the six clusters were due to intrinsic genetic factors, there would be high genetic variance between the clusters. The average mutation counts for the six clusters were 6.38, 3.49, 6.57, 7.09, 7.89, and 8.96 (Appendix A), respectively. Considering the different collection dates (Figure 3A) of the strains, mutation rates as opposed to mutation counts were more effective for describing the genetic variations between clusters. The average mutation rates for the six clusters were 25.55, 15.91, 25.44, 31.64, 30.99, and 34.12 substitutions per year, respectively. Specifically, the average mutation rate in cluster B was significantly lower (*p*-value < 0.001, Wilcoxon rank-sum test) than those in other clusters. In contrast, the average mutation rate in cluster F was significantly higher (*p*-value < 0.001, Wilcoxon rank-sum test) than those in other clusters. The Simpson’s diversity indexes for the six clusters were 0.7616, 0.7608, 0.8398, 0.8466, 0.8082 and 0.8502, respectively. Both the average mutation rate and Simpson’s index were highest in cluster F, suggesting that the diversity of cluster F was higher than the other clusters. The nucleotide diversity per site for the six clusters was 0.0196%, 0.0222%, 0.0171%, 0.0256%, 0.0131%, and 0.0132%. The high mutation rates but low nucleotide diversity in cluster E and cluster F suggest that these two clusters may have more fixed mutations than the other clusters. The nucleotide diversity values for each gene across all clusters are displayed in Figure 3B–G. Except for some short genes that were unlikely to be informative, the diversity of most genes was close to the diversity of their genome-wide variants. Our analysis showed that intra-cluster genetic diversity differed between clusters, suggesting that selective pressures might be different between clusters. These different selective pressures will affect the geographical distribution of each cluster.

### 3.4. Exploring the Mutations That Shaped the Geographical Distribution of Population Structure

The high genetic diversity between clusters indicated that the frequencies of the mutations across clusters were very different. In order to explore whether there are mutations that affect the genetic structure within the clusters, we applied ANOVA to identify the statistically significant mutations that were strongly associated with clusters. Across the 7970 substitutions, 26.27% (2094 substitutions) of them achieved *p*-values < 0.05 (Appendix A). We found that some of these mutations were fixed or nearly fixed (frequency > 95%) in one or several clusters (Figure 4A and Appendix A). Cluster C, cluster E and cluster F shared four common fixed substitutions: A23403G, C241T, C3037T and C14408T. Cluster E had two additional fixed substitutions: C1059T and G25563T. Cluster F had three additional substitutions from position 28,881 to position 28,883. For the remaining three clusters, there were two fixed substitutions (C8782T, T28144C) in cluster D and three fixed substitutions (G11083T, G14805T, and G26144T) in cluster A. It is noteworthy that the fixed mutation numbers in cluster E (six) and cluster F (seven) were higher than in any of the other clusters, which was consistent with our conclusion of the high mutation rates but low nucleotide diversity in cluster E and cluster F.

We selected the 2% (42/2094) substitutions that achieved the lowest *p*-values (Table 2) and analyzed their distributions in the clusters. Of the 42 substitutions, there were 26 nonsynonymous mutations (mutation G28882A was in a trinucleotide mutation from position 28,881 to 28,883, spanned two codons and resulted in an RG (arginine-glycine) to KR (lysine-arginine) amino acid change). We focused on these nonsynonymous mutations, as these mutations may comprise those that affect the population structure [31]. Indeed, many of these substitutions have been reported to impact the evolution of SARS-CoV-2 [32,33].

Previous studies have reported that recombination is common in coronavirus [3,32,33]. Given that recombinations in SARS-CoV-2 may perturb the clustering, we used Haploview [34] to analyze the linkage disequilibrium (LD) by calculating standardized disequilibrium coefficients (D′) and squared allele-frequency correlations (*r*^2^) of the 42 substitutions. D′ is affected solely by recombination and not by differences in allele frequencies between sites, and *r*^2^ is also affected by differences in allele frequencies at the two sites. In the heatmap of D’ and *r*^2^ (Appendix A), no obvious LD blocks were discovered, indicating that our clustering of SARS-CoV-2 strains using substitutions was not distorted by recombination.

Selection usually affects the distribution of the mutations in the population. Purifying selection tends to remove amino acid-altering mutations, while positive selection tends to increase the frequency of the mutations. Considering the rapidly proliferating pattern of SARS-CoV-2 that strengthened the power of drift relative to the power of purifying selection [35,36,37], we mainly focused on the positive selective sites. We applied HyPhy [38] to infer the probabilities of the extracted 26 nonsynonymous mutations under positive selection. There were nine mutations (asterisks in Table 2) with a positive probability >0.95. In particular, mutations G2891A, G11083T, C14408T, C17747T, and A23403G (D614G) were reported as recurrent mutations [32,39]. The recurrence of these mutations agrees with the assumption that they may confer selective advantages in the population. These possible positively selected cluster-specific mutations may result in higher genetic distances between clusters of SARS-CoV-2 across geographical regions.

### 3.5. The Global Spread of SARS-CoV-2

Regardless of genetic factors, the travel of humans can also lead to unique geographical distributions in today’s highly globalized world. By analyzing the frequencies of the extracted 42 mutations in each cluster (Figure 4A) and their collected daily counts (Figure 4B), we can trace the dynamics of the substitutions in the SARS-CoV-2 genome. The four genetically linked mutations—A23403G (D614G), C241T, C3037T, and C14408T—that were fixed across three clusters (C, E and F) were the highest frequency mutations in the world, with a high frequency on all continents in our downloaded sequences, including South America (87%), Africa (86%), Europe (75%), North America (65%), Oceania (55%), and Asia (32%). The earliest time when sequences carrying these mutations was collected was in late January 2020. About a month later, these mutations were discovered worldwide. Though the mutation A23403G (D614G) has been reported and estimated to be a positive selective mutation, it is almost impossible to spread across the world without human migration in such a short time. Besides these high-frequency mutations, some lower-frequency mutations also provided some evidence of human migration. We explored the geographical distributions of mutations with global frequencies < 0.05 in Table 2. Though most of these low frequency mutations were mainly collected within a single continent, we still found two mutations—T28688C and G1397A—that were discovered in Asia, Europe, and Oceania with high proportion. In addition, the spatial geographical distributions of some substitutions also provide evidence that human migration may have influenced the spread of the virus. For example, on the west coast of the USA, most strains accumulated the mutations C8782T and T28144C (cluster D), and these mutations were also found in high frequencies in East Asia. In contrast, on the east coast of the USA, most strains accumulated the mutations A23403G, C241T, C3037T, C14408T, C1059T, and G25563T (cluster E), and similar strains were mainly discovered in Europe (Appendix A). Note that, with the rapid growth of SARS-CoV-2, government policies were implemented, including travel bans, to restrict the spread of the virus. As a result, some mutations may have experienced weaker selective pressure by human intervention, which also affected the geographical distributions of SARS-CoV-2.

## 4. Discussion

Deciphering the genetic structure and subtypes of SARS-CoV-2 is critical for reducing the risk of future dissemination and assessing efficacy of antibodies on different strains [40]. In this study, we employed a clustering model along with phylogenetic analysis to illuminate the subtypes of 16,873 SARS-CoV-2 strains. Using the substitution profiles of the strains as input features, we identified six major clusters of the strains. To our surprise, we found specific geographical distributions of the clusters, most of which, except Oceania, were mainly composed of one to two clusters. We used complementary approaches to evaluate whether the geographical distributions for the clusters were due to genetic factors or human travel. The varied intra-cluster genetic diversity across the clusters suggested different selective pressures between clusters, which would affect the geographical distribution of the clusters. By analyzing the statistically significant mutations that were strongly associated with the clusters, we identified that some mutations might be under positive selection, indicating different geographical distributions between the clusters were partially affected by these mutations. In addition, the dynamics and the spatial geographical distributions of some substitutions suggested that human migration may also have affected the different geographical distributions. In general, our findings indicate that the geographical distributions of the clusters are the result of both genetic factors and the migration of humans.

One limitation of our study is the sampling bias of SARS-CoV-2 collections. Indeed, approximately 85% of the strains were from North America and Europe (Appendix A). In particular, the proportion of strains from the United Kingdom and the USA was over 60%. As a comparison, the summation of proportions from the two continents of Africa and South America was smaller than 2% (Appendix A). The sampling bias may lead to the underestimation of certain mutations that are frequent in continents with fewer collected strains. For example, the frequency of mutation C15324T reached 41.84% in Africa, but was only 2.21% outside Africa. The frequency of mutation T29148C reached 15.13% in South America, but was only 0.12% outside South America. Another mutation T27299C with a frequency of 10.92% in South America was only found with a frequency of 0.08% in other regions. In fact, all three mutations were mostly grouped in single clusters, indicating these mutations were highly concentrated. However, due to the small proportion of the strains from these two continents, these mutations were unable to affect the clustering of samples. To address this issue, more strains were needed to be collected from these continents. In addition, we found that in cluster B, there were no fixed mutations. We calculated the pairwise dependency scores (see Section 2) of all the mutations with frequencies > 0.05 in cluster B and discovered five main subclusters (Appendix A). Other than the mutation G11083T that was discovered in two subclusters, there were no common mutations between either of the five clusters. As shown in Figure 3A, strains in cluster B were grouped in one cluster mainly because these strains had smaller mutation counts than strains in other clusters. The genetic distance between two strains was still small, though they shared no common mutations. To address this issue, another clustering assessment can be used for further analyses. Moreover, considering that our clustering model assumed that every nucleotide had the same rate of changing into other nucleotides (JC69 [41]), this model is more suitable for fast-growing species within a short period. For long-term population clustering, the input features need to include a more complicated substitution model such as TN93 [42], GTR [43], and more.

Despite these limitations, our discovery of high genetic diversity in early SARS-CoV-2 is consistent with other published studies [44]. In addition, mapping the clustering results to the phylogenetic tree suggested that the emergence and the spread of SARS-CoV-2 occurred in short intervals, similar to previous studies [2,45,46,47]. Our identification of the main substitutions that drove the clustering results has also been reported by many studies concerned with the evolution of SARS-CoV-2. For example, mutation A23403G (D614G, Asparticacid to Glycine) in the *spike* protein domains was reported to show significant variation in cytopathic effects and viral load, and substantially change the pathogenicity of SARS-CoV-2 [48]. This mutation is accompanied by a mutation (T14408C) that results in an RNA-dependent RNA polymerase (RdRp) amino acid change [49]. In addition, Tang et al. [50] used mutation T28144C to define “L” types (defined as “L” type because T28,144 is in the codon of Leucine) and “S” types (defined as “S” type because C28,144 is in the codon of Serine) of SARS-CoV-2. They found that the “L” type was more transmissible and aggressive than the “S” type.

In summary, our current study, as well as previously published unsupervised clustering-based SARS-CoV-2 analyses [51,52], have showcased the advancement and efficiency of using clustering models in supporting phylogenetic analysis, especially when investigating the population structures of these never-seen-before species.

## 5. Conclusions

The SARS-CoV-2 virus has been spreading rapidly and across the globe since first being reported in December 2019. To understand the evolutionary trajectory of the coronavirus, phylogenetic analysis is needed to study the population structure of SARS-CoV-2. As sequencing data worldwide is accruing rapidly, grouping them into clusters helps to organize the landscape of population structures. To effectively group these data, we utilized clustering models to partition the viral sequences into clusters. The partition of the viral sequences revealed six major clusters, and these clusters differed in their geographical distributions. Through multiple approaches, we found that the unique geographical distributions across the clusters were influenced by both intrinsic genetic factors and the migration of humans. Identifying the mutations that are strongly associated with specific clusters has potential implications for the diagnosis and pathogenesis of COVID-19. In addition, our application of clustering techniques has proven to be a valuable method for studying the structure of new and unknown populations.

## Figures and Tables

**Figure 1 genes-13-00648-f001:**
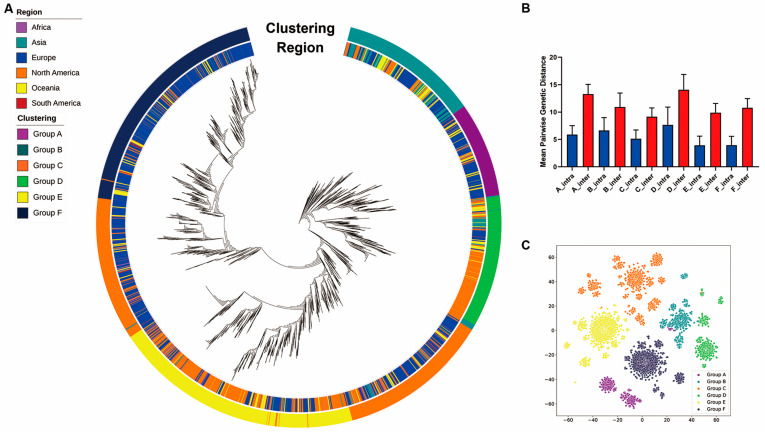
Clustering of SARS-CoV-2. (**A**) Phylogenetic tree of the 16,873 SARS-CoV-2 strains. The inner colored panel represents the continent for each collected strain, and the outer colored panel represents the partitions of the six clusters in the tree. (**B**) Mean intra-cluster and inter-cluster pairwise genetic distances across the six clusters. The blue bars represent mean pairwise genetic distances between pairs of isolates within the corresponding clusters, and the red bars represent mean pairwise genetic distances between pairs of isolates outside the corresponding clusters. The error bars display the standard deviations. For all six clusters, the mean intra-cluster pairwise genetic distances are significantly lower than the corresponding mean inter-cluster pairwise genetic distances (*p*-value < 0.001, Wilcoxon rank-sum test). (**C**) The t-SNE plot displays the clustering of the strains.

**Figure 2 genes-13-00648-f002:**
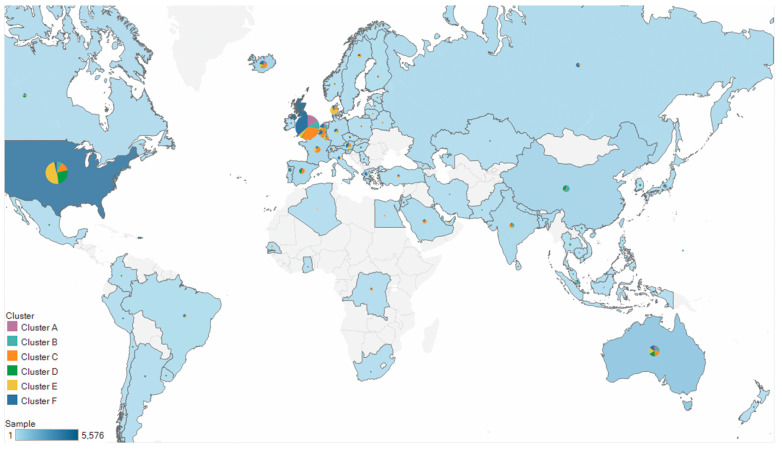
Geographic distributions of the six clusters. Pie charts display the proportions of six clusters among all SARS-CoV-2 strains in each country. Circle sizes and the color scales correspond to the number of strains analyzed per country.

**Figure 3 genes-13-00648-f003:**
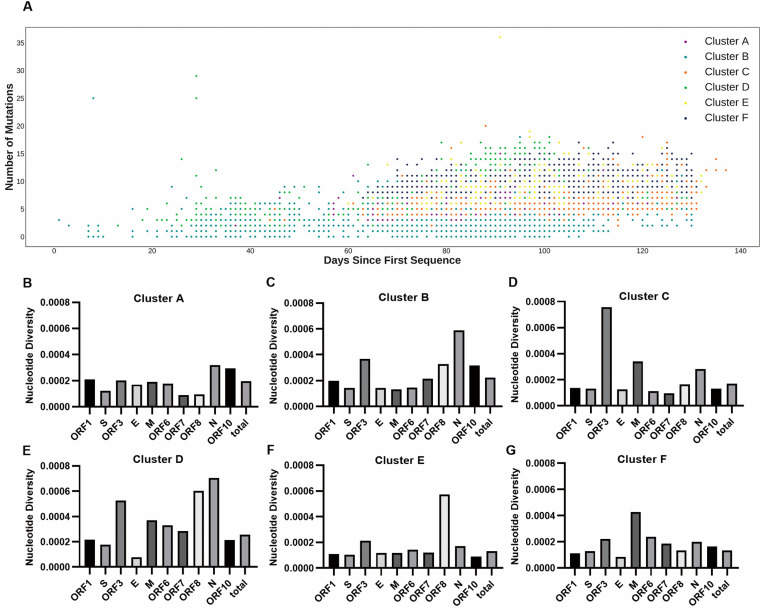
The genetic diversity between clusters. (**A**) The mutation counts over days of 16,873 SARS-CoV-2 strains. The X axis represents the days from the corresponding collection date of strains to 24 December 2019 when the earliest strain (EPI_ISL_402123) was collected. The Y axis represents the number of mutations of each collected strain. A mutation is defined by a nucleotide change from the original nucleotide in the reference genome to the alternative nucleotide in the studied viral genome. (**B**–**G**) The intra-cluster nucleotide diversity (π) per site for each gene and genome-wide across six clusters.

**Figure 4 genes-13-00648-f004:**
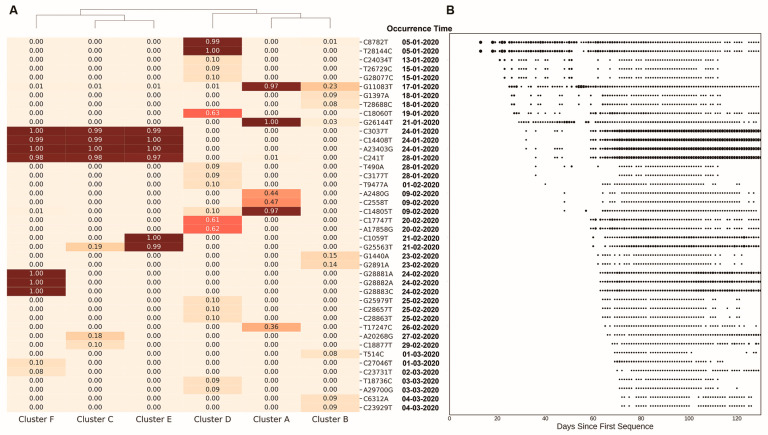
The clustering of the six clusters by the extracted mutations. (**A**) The heatmap displays mutation frequency of the 42 mutations across six clusters. The colors and values represent different frequencies of the corresponding mutations in each cluster. The collected days of the mutations are represented in (**B**). The X axis represents the days from the corresponding collection date of strains to 24 December 2019 when the earliest strain (EPI_ISL_402123) was collected. Circle sizes represent the frequency the of the mutations on each collection day.

**Table 1 genes-13-00648-t001:** Geographic distribution of six continents for each cluster.

Cluster	Cluster A	Cluster B	Cluster C	Cluster D	Cluster E	Cluster F	Total
Africa	3	4	65	7	10	9	98
Asia	38	648	248	217	57	116	1324
Europe	1137	990	3119	212	1108	2961	9527
North America	94	334	625	1268	2274	170	4765
Oceania	110	161	233	196	191	149	1040
South America	6	5	44	10	5	49	119
Total	1388	2142	4334	1910	3645	3454	16,873

**Table 2 genes-13-00648-t002:** The information of the 42 mutations using ANOVA.

Mutation	Substitution	Amino Acid Substitution	Type	Gene	Frequency	Cluster					
						A	B	C	D	E	F
C241T	C>T	Intron	Intron	Intron	66.37%	10	10	4238	2	3548	3391
T490A	T>A	D>E	N	ORF1ab	1.04%	0	0	1	174	0	0
T514C	T>C	H>H	S	ORF1ab	0.97%	0	162	1	0	0	0
C1059T *	C>T	T>I	N	ORF1ab	21.69%	1	8	2	0	3645	3
G1397A	G>A	V>I	N	ORF1ab	1.12%	0	186	0	0	1	2
G1440A	G>A	G>D	N	ORF1ab	1.92%	0	324	0	0	0	0
A2480G	A>G	I>V	N	ORF1ab	3.60%	608	0	0	0	0	0
C2558T	C>T	P>S	N	ORF1ab	3.83%	646	1	0	0	0	0
G2891A *	G>A	A>T	N	ORF1ab	1.77%	0	298	0	0	0	0
C3037T	C>T	F>F	S	ORF1ab	67.26%	2	7	4277	3	3611	3448
C3177T	C>T	P>L	N	ORF1ab	1.05%	0	0	1	171	6	0
C6312A	C>A	T>K	N	ORF1ab	1.14%	0	189	1	0	0	3
C8782T	C>T	S>S	S	ORF1ab	11.42%	1	21	5	1898	1	1
T9477A	T>A	F>Y	N	ORF1ab	1.17%	0	3	0	195	0	0
G11083T *	G>T	L>F	N	ORF1ab	11.81%	1342	485	52	21	54	39
C14408T *	C>T	P>L	N	ORF1ab	67.47%	1	8	4301	2	3636	3436
C14805T	C>T	Y>Y	S	ORF1ab	9.39%	1352	8	1	195	0	28
T17247C	T>C	R>R	S	ORF1ab	3.00%	500	5	1	0	0	0
C17747T *	C>T	P>L	N	ORF1ab	6.92%	1	0	0	1165	1	0
A17858G	A>G	Y>C	N	ORF1ab	7.05%	1	1	0	1187	0	0
C18060T	C>T	L>L	S	ORF1ab	7.16%	0	3	2	1202	1	0
T18736C	T>C	F>L	N	ORF1ab	1.01%	0	0	1	169	0	0
C18877T	C>T	L>L	S	ORF1ab	2.67%	2	2	440	4	0	2
A20268G	A>G	L>L	S	ORF1ab	4.61%	0	1	773	3	0	1
A23403G *	A>G	D>G	N	S	67.65%	4	4	4316	6	3634	3451
C23731T	C>T	T>T	S	S	1.68%	0	0	0	0	1	282
C23929T	C>T	Y>Y	S	S	1.13%	0	186	1	0	1	2
C24034T	C>T	N>N	S	S	1.16%	0	2	1	187	4	1
G25563T *	G>T	Q>H	N	ORF3a	26.44%	1	3	829	2	3625	2
G25979T	G>T	G>V	N	ORF3a	1.16%	0	2	1	193	0	0
G26144T *	G>T	G>V	N	ORF3a	8.61%	1387	62	0	1	1	1
T26729C	T>C	A>A	S	M	1.07%	0	1	1	179	0	0
C27046T	C>T	T>M	N	M	2.13%	0	1	5	0	0	353
G28077C	G>C	V>L	N	ORF8	1.13%	0	1	1	188	0	0
T28144C *	T>C	L>S	N	ORF8	11.36%	0	10	1	1903	2	0
C28657T	C>T	D>D	S	N	1.21%	0	3	3	196	1	2
T28688C	T>C	L>L	S	N	1.07%	0	178	1	0	1	0
C28863T	C>T	S>L	N	N	1.19%	1	2	2	193	2	0
G28881A	G>A	R>K	N	N	20.54%	4	3	3	1	1	3453
G28882A	G>A	R>K ^1^	N	N	20.49%	1	2	0	0	0	3454
G28883C	G>C	G>R	N	N	20.49%	1	2	1	0	0	3453
A29700G	A>G	Intron	Intron	Intron	1.04%	0	0	4	167	4	1

^1^ G28881A and G28882A occur within the same codon. Amino acid annotation (R>K) is based on the co-occurrence of these mutations. * Under positive selection inferred by HyPhy.

## Data Availability

The publicly available SARS-CoV-2 genomes in this study are available at GISAID (https://www.gisaid.org, accessed on 15 May 2020). The reference SARS-CoV-2 genome is available at the NCBI GenBank (GenBank Accession Number: NC_045512.2, https://www.ncbi.nlm.nih.gov/nuccore/NC_045512.2, accessed on 15 May 2020).

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
