# Peer review of "Using an Unsupervised Clustering Model to Detect the Early Spread of SARS-CoV-2 Worldwide"

_genes, 2022, doi:10.3390/genes13040648_

Round 1

Reviewer 1 Report

In the presented manuscript Li et al. developed a clustering algorithm which uses a machine learning innovation and applied this clustering algorithm to analyse a large dataset of full genome sequences of SARS-CoV-2 sequences. Additionally, authors characterised genetic and evolutionary features of these phylogenetic clusters, and interrogated their geographical distribution. Overall, the manuscript shows an interesting methodology and most of the manuscript is well presented.

I have following suggestions for the authors:

  • In the Discussion, please comment whether this methodology can be used for other viruses, including newly emerging strains. If yes, please briefly (one sentence) discuss limitation of the method when applied to other viral genera.
  • Please give examples of other phylogenetic methods which use machine learning; this could be added to the Introduction or Discussion.
  • Ln 40 – 42: Can you briefly elaborate why pairwise distance is not sufficient?
  • Reference 25: is this a publication in a peer-reviewed journal or conference notes? If first: how is this different to the presented manuscript (the reviewer couldn’t access this reference), if second: only peer-reviewed manuscript should be placed in the list of references.
  • 1A: Please improve colouring contrast in the figure, e.g., colouring used for Africa and North America cannot be distinguished.
  • 1: legend contains section E and F, while figure B and C.
  • 1B and Ln 188-9: please provide details of inter-cluster pairwise distance: did all isolates from a cluster were compared to all isolates from other clusters, or was it based on a subsample of isolates. If latter, please explain the strategy for subsample selection.
  • 2: please provide a higher resolution figure, small pie charts are invisible upon zoom-in.
  • Fig 3A: is it possible to include a 3rd axis which will represent genomic position? If yes, please revise the figure accordingly.
  • Figure 3, Ln 250: should be “The nucleotide intra-cluster genetic diversity ..”
  • Ln 304-5: please specify if the mutations authors refer to are cluster specific.
  • Result section 3.4 and 3.5 could be made more concise. In these sections the focus of the manuscript is lost. Ln 272-81 belong to discussion.
  • Ln 258-66: could benefit from a graphical representation of the shared mutations.
  • Ln 344-6: this sentence is incomplete.
  • Ln 370-2: belongs to the Results.
  • Ln 374-5: Please specify what means “these strains”.

Author Response

Dear Editors and Reviewers:

Thanks very much for sending us your valuable comments and criticisms on our manuscript, which are extremely helpful in improving our manuscript. Based on the reviewers’ recommendations, careful modifications have been made to our manuscript. The following is a detailed list of point-by-point responses to all comments (Please see the attachment). With these changes, we hope it is now acceptable for publication in “Genes”.

Thank you and best regards.

Yours sincerely,

Yuan Luo

Reviewer 2 Report

Summary

With the continuously accruing of SARS-CoV-2 genomic sequences, traditional pair-wise distance based phylogenetic tree approach faces challenges in handling the large number of genomes. The authors tried to use a previously developed, unsupervised clustering approach to group a total 16873 SARS-CoV-2 genomes from the early stage of spreading.

The authors used 16873 genomes that were download from the CISAID database and passed their quality control. Sequences from Europe and North America made up the majority of the samples (9527+4765 out of 16873). Previous publication using the same dataset revealed 6 major clusters. The authors mapped these clusters agains the pylogenetic tree and found the clusters largely matches the tree groups, with the exception of cluster C. Different geographical distribution of these 6 clusters were observed. The author then tried to evaluate the contribution from intrinsic genetic variance and human migration to the difference in observed geographic distribution across identified 6 clusters.

While the overall goal may carry significant clinical importance, the claims by the authors were not well supported by convincing evidence from the results. The findings merely add to our current understanding of the early spreading of SARS-CoV-2, nor do the methods demonstrate superiority in handling the increasing amount of genomes data. See the detailed comments below.

Major:

  1. This manuscript seems to significantly overlap with the authors’ previous publication at IEEE conference (referecen #25)
  2. Line 22-23: “This study provides a concrete framework for using clustering methods to study the population structure of a never-seen-before species like SARS-CoV-2” is overstated.
  3. Since the deposit of SARS-Cov-2 genomes is fastly increasing daily, for the accuracy and reproducibility of these results, it is critical that the author states the date of data download and provides the accession numbers of the genomes used in this study. The author should also specify the time window of the data coverage to better define “early spread”
  4. Line 178, how the phylogenic tree was built/obtained needs to be described. Line 180: Methods for calculating pairwise genetic distance between intra clustering and inter cluster were not describe, nor did the tSNE embedding. Without such essential details, it is challenging to justify the results presented.
  5. Results from figure 1A and 1C both suggest that cluster C is should be further broken into subclusters. Thus the statement in line 181-182 is inaccurate. I wonder if those subclusters correspond to geographic groups. It would be also informative if the author could color the tSNE plot by geographic groups. In addition, since the DEC model requires manual specification of the number of clusters which is critical for the downstream analysis, the authors should provide the supporting evidence for optimal cluster number selection.
  6. Line 198: “Asian and North American strains occupied 76.47% 198 in cluster B.” is incorrect. Those two make up ~45.8% of cluster B. In addition, it is misleading to call the strains by their geographic locations, such as European strains.
  7. During the early stages of spreading, travel bans played important role in restricting the spreading of the virus. Grouping the geographic distribution by continent without considering the travel restrictions is too simplified and could be misleading.
  8. Line 227-228:Definition of mutation rate was not given
  9. The authors’ approaches in demonstrating the high genetic variance between clusters are not convincing. Mutation rate comparison across clusters can not accurately indicate genetic variations. The difference in the mutation rates and Simpson’s indexes were small and are challenging to accurately and quantitively compare the genetic variance with confidence. Furthermore, the authors’ statement “suggesting that selective pressures were different between clusters” was not clearly supported by the data. Other very likely factors, such as traveling, were not considered when making such statement in line 241
  10. The results from the ANOVA analysis on the 7970 substitutions (line 255-265) were not included in the results, other than the authors’ description of a slection of fixed substitutions. The author did not provide a summary of those fixed substitutions, which made it challenging to evaluate the conclusions made on the fixed mutations in clusters E and F (line 564-266).
  11. It is unclear why the author picked 2% as the selection cutoff in line 267. Will this arbitory selection biase the results? Authors please comment
  12. Nonsynonymous mutations could be both under positive and negative selections. The selection depends on the nature of the mutation. Thus “26 nonsynonymous mutations that were under positive selection” in line 266-267 is incorrect.
  13. The conclusion in line 304-305 is not well supported and is confusing to follow, because normally positive selection is thought to reduce diversity.
  14. How do the clusters learn in this study, and the cluster-specific mutations match the variants annotations that had been used in tracking the spreading?
  15. Travel-based spreading of this virus is a widely accepted fact. The authors’ effort in section 3.5 does not add novel insights into understanding the early spreading.

Minor:

Singletons and doubletons in line 162 should be defined.

Figure 1A the clustering legend color does not match the plot (group C)

Line 159: “6.99 per mutation count per strain”..==> “6.99 mutation count per strain”

Line 262-263, it is confusing why the author stated “two fixed substitutions (C8782T, 262 T28144C)” and “three fixed substitutions (G11083T, G14805T and G26144T)” separately instead of saying a total of 5 fixed substitutions.

Author Response

(The authors gave the same response as above.)

Reviewer 3 Report

This paper provides an interesting way to use clustering of viral (SARS-CoV-2) sequences to support phylogenetic analysis, and uncovers some interesting trends of SARS-CoV-2 dynamics

One general comment is that, while this is not necessary, was there ever a comparison of your results to known variants and their global distribution from other studies?  For example, the GISAID resource itself has "variant surveillance" metadata with "pango" lineage identifiers, e.g., Alpha = B.1.1.7, etc.  Even a comparison of your results to this metadata could be interesting

I only have the following remaining (minor) comments:

p.1 line 34: "with high efficiency"

p.1 line 39: "A phylogenetic tree"

p.1 line 41/42: I think this is described in more detail in the main text, but it is a bit unclear initially what "using pairwise distances on the phylogenetic tree fails to intuitively depict the entire structure of the population" means.  Do you mean that this information alone (without further processing) does not intuitively depict the entire structure?  Because, ultimately, it is this information that is used as the source (after some processing) to build the phylogeny, the clustering, etc.  Maybe a sentence elaborating the intended meaning of the above phrase could help

p.2 line 68: "algorithms to assist phylogenetic analysis"

p.2 line 69: "across humans worldwide"

p.3 line 116: "dimensionality reduction"

p.4 line 174: Do you have a citation to back-up the claim that SARS-CoV-2 undergoes ongoing horizontal gene exchange?

Figure 1 caption: I think (E) and (F) should be (B) and (C) ?

Author Response

(The authors gave the same response as above.)

Round 2

Reviewer 2 Report

the authors have reasonably addressed all my questions